# 4-Substituted Pyridine-3-Sulfonamides as Carbonic Anhydrase Inhibitors Modified by Click Tailing: Synthesis, Activity, and Docking Studies

**DOI:** 10.3390/ijms26083817

**Published:** 2025-04-17

**Authors:** Krzysztof Szafrański, Jarosław Sławiński, Anna Kawiak, Jarosław Chojnacki, Michał Kosno, Andrea Ammara, Claudiu T. Supuran

**Affiliations:** 1Department of Organic Chemistry, Medical University of Gdańsk, Al. Gen. J. Hallera 107, 80-416 Gdańsk, Poland; 2Intercollegiate Faculty of Biotechnology, University of Gdańsk, Abrahama 58, 80-307 Gdańsk, Poland; 3Department of Inorganic Chemistry, Faculty of Chemistry, Gdańsk University of Technology, G. Narutowicza 11/12, 80-233 Gdańsk, Poland; 42nd Department of Radiology, Medical University of Gdańsk, Mariana Smoluchowskiego 17, 80-210 Gdańsk, Poland; 5Section of Pharmaceutical and Nutraceutical Sciences, Department of NEUROFARBA, University of Florence, Polo Scientifico, Via U. Schiff 6, Sesto Fiorentino, 50019 Firenze, Italy

**Keywords:** carbonic anhydrase, CuAAC, sulfonamide, 1,2,3-triazole, molecular docking, anticancer screening

## Abstract

In the search for new selective inhibitors of human carbonic anhydrase (hCA), particularly the cancer-associated isoforms hCA IX and hCA XII, a series of 4-substituted pyridine-3-sulfonamides was synthesized using the “click” CuAAC reaction, proven by X-ray crystallography, and evaluated for their inhibitory activity against hCA I, hCA II, hCA IX, and hCA XII. Additional molecular docking studies and cytostatic activity assays on three cancer cell lines were conducted. The compounds exhibited a broad range of inhibitory activity, with K_I_ reaching 271 nM for hCA II, 137 nM for hCA IX, and 91 nM for hCA XII. Notably, compound **4** demonstrated up to 5.9-fold selectivity toward the cancer-associated hCA IX over the ubiquitous hCA II, while compound **6** exhibited a remarkable 23.3-fold selectivity between transmembrane isoforms hCA IX and hCA XII. Molecular docking studies have shown the possibility of selective interaction with the hydrophilic or lipophilic half of the active site, what results from the adjacent (3,4) position of the “tail” in relation to the sulfonamide group.

## 1. Introduction

Carbonic anhydrases (CAs, EC 4.2.1.1) form a diverse group of metalloenzymes present in both prokaryotic and eukaryotic organisms. These enzymes play a pivotal role in physiological processes by facilitating the reversible conversion of carbon dioxide into bicarbonate ions and protons. In humans, CAs are represented by 16 isoforms of the α-class, each exhibiting distinct cellular localization, tissue specificity, and functional properties. Based on their location within cells, these isoforms are categorized as cytosolic, mitochondrial, membrane-associated, or secreted, reflecting their broad involvement in metabolic and regulatory pathways [1]. Among them the membrane-associated isoforms CAIX and CAXII are particularly significant in the context of oncology. CAIX is frequently overexpressed in hypoxic tumor microenvironments, where it enables cancer cells to adapt to low oxygen levels and maintain intracellular pH balance. This activity supports processes such as tumor invasion, migration, and resistance to therapies. Also, the CAXII isoform has been linked to cancer progression in certain tumor types, including breast and kidney cancers. Notably, the expression of CAIX and CAXII is limited in normal tissues, making them compelling targets for anticancer therapies.

Selective inhibition of carbonic anhydrase activity has been shown to exert anticancer effects [2]. One of the most prominent examples is the sulfonamide compound SLC-0111, which specifically targets the activity of carbonic anhydrase IX. Several preclinical studies have demonstrated the antitumor efficacy of SLC-0111 across various solid tumor models, including triple-negative breast cancer, pancreatic cancer, glioblastoma, and melanoma, leading to the advancement of SLC-0111 into Phase I clinical trials [3]. In addition to their direct effect on tumor tissue, the efficacy of combining human carbonic anhydrase inhibitors with other chemotherapeutic agents in cancer therapies has been well-documented [4]. Recent studies have also shown that CAIX interacts with multiple signaling pathways involved in the cellular response to radiation, which led to the finding that CAIX inhibitors may exert a synergistic effect by enhancing tumor radiosensitivity when used in combination with radiotherapy [5]. Similarly, the conjugation of CAIX targeting ligands with photothermal agents has been employed in photothermal therapy for cancer treatment. This approach allows for the selective targeting of hypoxic regions within tumors, thereby improving drug accumulation at the tumor site and increasing therapeutic efficacy [6]. Furthermore, recent studies suggest that CAIX inhibition can sensitize cancer cells to oxidative stress and iron-dependent cell death mechanisms, such as ferroptosis and ferroapoptosis, highlighting its potential role in novel combination therapies targeting tumor metabolism and redox balance [7,8].

Sulfonamides are one of the most effective classes of carbonic anhydrase inhibitors, acting by specifically binding to the active site of the enzyme. The mechanism involves the sulfonamide group in anionic form (SO_2_NH^−^) interacting directly with the zinc ion coordinated by another three histidine residues, replacing the zinc-bound water or hydroxide ion. This interaction disrupts the enzyme’s ability to facilitate the reversible hydration of carbon dioxide, effectively inhibiting its activity. Over the years, sulfonamide-based inhibitors have been widely used as diuretics or in the treatment of glaucoma and epilepsy. Despite their potency, many sulfonamide inhibitors lack isoform selectivity, leading to off-target effects. Due to the high structural similarity of the active sites across various human CA isoforms, the most commonly employed strategy for CA inhibitor selectivity is the so-called “tail approach”. In this approach, a zinc-binding group, typically a sulfonamide moiety attached to an aromatic (typically a phenyl) ring, facilitates non-selective binding to the zinc cation. Meanwhile, a long chain on the opposite side of the aryl ring interacts with amino acids at the entrance to the active site, conferring selectivity [9,10]. This method, improved even by introducing more than one tail, has yielded many selective inhibitors [11,12,13]. On the other hand, the other method used in the design of CA inhibitors is the “ring approach”, which is focused on modifying the scaffold attached to the zinc-binding group (ZBG). Techniques such as perfluorination of the aromatic ring, which increases the acidity of the sulfonamide nitrogen and thus enhances the ionized, active form of the sulfonamide group, or the use of five-membered heterocyclic rings instead of benzene sulfonamide, have been explored. The most significant success with this approach has been the modification that enabled the development of water-soluble dorzolamide [14], used topically to treat glaucoma by inhibiting carbonic anhydrase isoform II directly in the eyeball.

The presented study focuses on the combination of these two approaches by utilizing pyridine-3-sulfonamide as the starting scaffold and exploring the variability in substituents at its position 4. According to the ring approach concept, the electron-withdrawing nature of the pyridine ring significantly increases the acidity of pyridine-3-sulfonamide compared to benzenesulfonamide. Whereas its susceptibility to nucleophilic aromatic substitution allows for the introduction of a wide range of substituents at position 4 of the pyridine ring, providing a large array of derivatives sourced from the same 4-chlorobenzene-3-sulfonamide starting compound. To date, our team has synthesized several dozen 4-substituted pyridine-3-sulfonamide derivatives [15,16,17] with nanomolar activity against hCA IX and hCA XII, demonstrating high selectivity—up to 50-fold—compared to hCA II (Figure 1). Another characteristic feature of these derivatives is the 3,4-substitution pattern relative to the sulfonamide group, which is analogous to *ortho* substitution in benzenesulfonamide derivatives. While benzenesulfonamides substituted with a “tail” in the *ortho* position are often considered less effective than those substituted in the *para* position, our studies, along with others [18,19,20,21,22], have shown that selecting the appropriate size of the tail substituent can yield compounds that are not only active but also highly selective for certain isoforms (Figure 2).

Considering the use of fast and universal tail modification techniques in the presented studies, we also incorporated commonly used in medicinal chemistry copper(I)-catalyzed azide-alkyne cycloaddition reaction (CuAAC) so-called “click chemistry” methods, which allow for the effective introduction of substituents with a wide structural variability. This method of so-called “click tailing” was also found useful in the modification of CA inhibitors, allowing for tail structure modifications [23], including the formation of glycoconjugates [24,25]. Our goal was to synthesize new compounds containing, at the 4-position of the pyridine sulfonamide ring, both small 1,2,3-triazole substituents, which could potentially interact closer to the active site, and long-chain substituents that would interact with more distant amino acids near the enzyme’s active site entrance.

## 2. Results and Discussion

### 2.1. Chemistry

The first set of 4-(4-R^1^-1,2,3-triazol-1-yl)pyridine-3-sulfonamides (**3**–**12**) was obtained from 4-azidopyridine-3-sulfonamide (**2**) prepared according to the previously described method by reacting 4-chloropyridine-3-sulfonamide with sodium azide. The CuAAC reaction of azide (**2**) and the appropriate terminal alkynes was carried out with copper(I) iodide as a catalyst in the presence of trimethylamine as the necessary amine ligand. The reaction was carried out in anhydrous acetonitrile at room temperature to obtain pure compounds **3**–**12** with a yield of 29–65% (Figure 1).

In the aromatic nucleophilic substitution reaction of **1** with propargylamine, we obtained compound **13** bearing a terminal alkyne moiety. On the other hand, derivative **14 [16]** containing a sulfur linker that was synthesized via thiouronium hydrochloride **14A** obtained by reacting **1** with thiourea and further treatment with propargyl bromide leading to the formation of a sulfide bond of compound **14** (Figure 2). Due to the worse solubility, the CuAAC reaction with alkynes **13** and **14** required changing the solvent to DMSO and using copper (II) sulfate CuSO_4_, reduced in situ with sodium ascorbate, as a catalyst. This synthetic route led to a “long-chain” series of compounds **15**–**23** with a yield ranging from 25 to 81% (Figure 2).

Despite the advantages of the CuAAC method, such as mild conditions and stereospecificity of the reaction, in the case of our compounds, we noticed an unfavorable tendency to form very stable, sparingly soluble complexes involving a pyridine ring with copper ions. This increased the difficulty of isolation, which in consequence negatively affected the reaction yield. 

### 2.2. Crystallographic Studies

To confirm the structure of the obtained compounds, especially the 1,4-substitution pattern of 1,2,3-triazole rings, X-ray crystallography studies of representative compounds **8** and **18** were carried out.

Solid **8** forms transparent needle crystals satisfying the symmetry of the triclinic system, the space group $P\bar{1}$ (no. 2). The molecular structure and atom labeling scheme are given in Figure 3. The crystal data, data collection, and structure refinement details are summarized in Appendix A. The asymmetric unit of compound **8** contains four molecules, and the whole unit cell contains eight molecules of the sulfonamide, *Z* = 8. The main difference in the four symmetry-independent molecules is in the various mutual orientations of the ring planes. Let us call the molecules A, B, C, and D following the sulfur atom labeling S1, S2, S3, and S4 (Figure 4). Two conformers, A and B, have the nitrogen-rich part of the triazole ring directed toward the sulfonamide group, and the other two, C and D, have antiparallel orientation of the moieties. Conformers A and B form intramolecular N-H…N (triazol) hydrogen bonding, stabilizing the mentioned conformations. Conformers C and D stabilize their conformation probably by the formation of weak (triazol)C-H…O (sulfonamide) hydrogen bonds. Obviously, there are also some intermolecular interactions that stabilize them in the solid state. In the crystals of **8**, molecules are linked also by intermolecular N-H…N (pyridine or triazol) or N-H…O interactions (see Appendix A). Most of the bond lengths and valence angles are in the expected ranges, and the differences between conformers A-D are not large. The three aromatic rings form an almost flat system only in the case of molecule D.

Compound **18** forms transparent needle crystals satisfying the symmetry of the monoclinic system, the space group $P2_{1}/c$ (no. 14). The asymmetric unit contains one molecule, and the whole unit cell contains four molecules of the sulfonamide, *Z* = 4. Most of the bond lengths and angles are in the expected ranges. The crystal data, data collection, and structure refinement details are summarized in Appendix A. The molecular structure and atom labeling scheme are given in Figure 5. The sulfonamide group is not ionized (deprotonated) in the solid state. Angles inside the 5-membered ring are close to the values expected for a regular pentagon of 3π/5 = 108°, with the smallest value at C8. The –NH_2_ group behaves as the hydrogen bond donor toward the O-atom from a sulfonamide group and the N-atom from a pyridine residue from two neighbor molecules in the structure (see Appendix A), which allows the formation of an infinite set of rings forming 2D layers propagating parallel to the crystallographic plane of Miller indices (100) (details on the hydrogen bonding are presented in Appendix A). Thus, hydrophobic and hydrophilic strata are present in the structure.

### 2.3. Carbonic Anhydrase Inhibition

The compounds **3**–**12** and **15**–**23**, as well as the standard, clinically used CA inhibitor acetazolamide **AAZ**, have been tested via a stopped-flow inhibition assay for the inhibition of four of the human-origin carbonic anhydrase isozymes, that is, two cytosolic, ubiquitous isozymes, hCA I and II, and two transmembrane (cancer-associated) isoforms, hCA IX and hCA XII. The results—inhibition constants K_I_ and selectivity index SI—are shown in Table 1.

The 1,2,3-triazole derivatives **3**–**12** and **15**–**23** demonstrate a lower inhibitory activity than that of the clinically used carbonic anhydrase inhibitor acetazolamide. In general, the mean inhibition constant K_I_ values for these compounds were lowest for hCA XII (mean = 1018 nM), followed by hCA IX (2502 nM) and hCA II (3808 nM). The highest K_I_ values, reflecting the lowest activity, exceeding 10,000 nM for all new compounds, were observed for the hCA I isoform, precluding the possibility of establishing a meaningful structure–activity relationship for this isoform. Yet, the following conclusions should be noted regarding the CA inhibitory data:I.Most compounds exhibited a similar or only slightly varied (compounds **4** and **7**) rank order of potency (K_I-CAXII_ < K_I-CAIX_ < K_I-CAII_), except for pairs **5**–**6** and **15**–**16**. Moreover, a statistically significant correlation can be observed between the activity against hCA II and hCA IX, with a Pearson correlation coefficient of 0.92.II.The inhibitory activity against the ubiquitous and fast-reacting isoform hCA II showed the greatest variability, ranging from 271.5 nM for compound **5** to over 10,000 nM for compounds **12**, **21**, and **22**. Among the derivatives **3**–**12**, in which the triazole ring is directly linked to pyridine, those containing a phenyl R substituent **8**–**12** were markedly less active, with potency further diminishing as the bulk and number of substituents increased. This trend was also observed across other tested isoforms. Conversely, aliphatic lipophilic substituents, such as n-hexyl (**5**) and 3-methylbutan-1-yl (**6**), showed the highest activity against hCA II in this series. A similar preference for aliphatic lipophilic substituents was observed in the **15**–**23** series, where compounds **17**, **18**, and **23** demonstrated K_I_ values of 419 nM, 505 nM, and 709 nM, respectively.III.The activity against hCA IX ranged from 137.5 nM for compound **4** to 8154 nM for compound **22**. In the **3**–**12** series, contrary to the trend observed for hCA II, the presence of aliphatic R^1^ substituents in compounds **5** and **6** significantly reduced activity compared to compounds **3**, **4**, and **7**. All compounds in the **15**–**23** series were more active against hCA IX than hCA II, with the highest activity observed for compounds **17** and **18** (R^2^ = cyclohexyl) and compound **23** (R^2^ = methyl-1,3-dioxolane).IV.The most sensitive isoform, hCA XII, was inhibited at concentrations as low as 91 nM by compounds **6**, **18**, and **23**, while compound **16**, the least active, exhibited a K_I_ of 4284 nM. Similar to hCA II, a positive impact of aliphatic R^1^ substituents in compounds **5** and **6** was observed, with these compounds displaying greater activity than compound **4**, which has an amide R^1^ substituent, and compound **7**, which has an alcohol R^1^ substituent.V.For the derivatives **15**–**22**, the nature of either the thiomethyl (-S-CH_2_-) or aminomethyl (-NH-CH_2_-) linker did not appear to critically influence activity, particularly against hCA II, as pairs of compounds with the same R^2^ exhibited similar activity profiles (refer to the heatmap in the table). For other isoforms, the effect of swapping sulfur and nitrogen is challenging to predict. For instance, in hCA XII, changing X from N to S can either halve the K_I_ (compounds **17**–**18** and **19**–**20**) or double it (compounds **15**–**16** and **21**–**22**), whereas in hCA IX, the same substitution yields the opposite effect (pairs **17**–**18** and **19**–**20**). The modification of the R^2^ substituent had a more pronounced effect.VI.In terms of selectivity toward tumor-associated isoforms IX and XII, it is noteworthy that compounds **4**, **7**, **15**, **16**, and **23** exhibited moderate (3 to 5.9-fold) selectivity for isoform hCA IX over isoform hCA II. The new compounds demonstrated significantly higher selectivity for hCA XII, with compounds **20**, **8**, **10**, and **9** exhibiting K_I_ values 9.1 to 18.7-times lower than those for hCA II.VII.Selectivity between the two tumor-associated isoforms is also significant, with compounds **6** (23.3-fold), **5** (18.2-fold), **20** (10.5-fold), and **8** (9.3-fold) favoring hCA XII, while compounds **16** (8.0-fold), **4** (4.4-fold), and **15** (4.0-fold) showed preference for hCA IX.

### 2.4. Molecular Docking Studies

Human carbonic anhydrase isozymes II, IX, and XII have relatively similar active sites, with differences primarily in a few amino acids. One of the key differences, which is often the molecular basis for the selectivity of the inhibitor, is the substitution of a critical amino acid at the entrance to the enzyme’s active site, where the bulky Phe131 in hCA II is replaced by valine in hCA IX or alanine in hCA XII, and as a result, significantly reducing the entrance volume of the hCA II binding pocket. Furthermore, the carbonic anhydrase binding pocket is known to be divided into two regions: a lipophilic formed by Ile91, Phe131, Val121, Val135, Leu141, Val143, Leu198, Pro202, Leu204 Val207, and Trp209 (CA II numbering) and a more hydrophilic region formed by Asn62, His64, Asn67, and Gln92 [26]; however, slight structural changes like modification Ala65 and Val135 to the serine residue in hCA XII affect and slightly differ its lipophilicity toward different isoforms [2,27].

To investigate the factors contributing to the variability in compound activity, we conducted molecular docking studies of compounds **3**–**12** and **15**–**22** into the active sites of hCA II (PDB code 6RH4), hCA IX (PDB code 5DVX), and hCA XII (PDB code 6QNG). To ensure accurate prediction of the binding poses in a wide range of structurally diverse compounds, the crystallographic structures of the enzymes were selected through extensive cross-validation, which involved redocking multiple (10–27) inhibitors, particularly 1,2-substituted arylsulfonamides from known crystallographic structures. The mean RMSD values of the top-scoring poses were as follows: for 6RH4, 2.07 Å (1.25 Å for the best-fitted but not best-scored) (Appendix A); for 5DVX, 3.63 Å (1.61 Å for the best-fitted) (Appendix A), and for 6QNG, 3.02 Å (1.64 Å for the best-fitted) (Appendix A). The calculations were performed using Molecular Operating Environment (MOE), 2022.02 software [28].

#### 2.4.1. hCA II

According to the calculations, while binding to the hCA II active site, all compounds exhibit the typical sulfonamide hydrogen bond with Thr199, which stabilizes the ionic bond between the sulfonamide anion and the zinc atom. However, comparing to the original ligand (4-nitrobenzenesulfonamide) of 6RH4, compounds display a different aryl ring orientation, i.e., the pyridinesulfonamide ring is twisted approximately 45 degrees, oriented laterally between the Leu198 and Val121 sidechains so that the pyridine nitrogen atom lies below the phenyl ring of Phe131, being an acceptor for a weak H_Ar_ hydrogen bond. The 1,2,3-triazole and R^1^ substituent for compounds **3**–**12** are directed toward the hydrophilic site of the binding cleft and pocket formed by Tyr7, His64, Ala65, His96, and Asn244. The 1,2,3 triazole ring of compounds **4**–**7** form Ar–Ar interactions with His94, while the long aliphatic chain of the most active compounds, **5** and **6**, aligns along the edge of Trp5, facilitating hydrophobic interactions (Figure 6). On the other hand, the carbonyl oxygen of **3** and **4** or the hydroxyl group of **7** may act as an acceptor for the hydrogen bond from the Asn62 amide sidechain.

More spacious and less labile phenyl R^1^ substituents in compound **8** and the addition of the next substituent in **9**–**12** push the pyridinesulfonamide ring toward the opposite side of the pocket, causing undesirable steric interactions.

The presence of a labile aminomethyl or thiomethyl linker between the pyridine and triazole rings allows compounds **15**–**23** to orient the R^2^ substituent toward the pocket entrance. The position of the substituent is stabilized by the interactions of triazole nitrogens with Trp5, Asn62, and His64. Additionally, the sulfur atom in the thiomethyl linker can act as an acceptor of the hydrogen bond from the hydroxyl group of Thr200 (Figure 6).

#### 2.4.2. hCA IX

In the hCA IX model, the replacement of Phe131 with a significantly smaller valine allows all compounds to adopt a different conformation, rotated by 180 degrees compared to hCA II, with the 1,2,3-triazole substituent oriented toward the lipophilic part of the active cleft, while the pyridine is directed with its nitrogen atom toward the hydrophilic region. This enables the formation of π–π stacking interactions with the pyrydinesulfonamide scaffold and His94, and arene–H interactions with the hydroxyl moiety of Thr200. The sulfonamide group’s anion, which complexes with the Zn^2+^ cation, is further stabilized by typical hydrogen bonds with Thr199. Among the small-substituent compounds (**3**–**7**), the oxygen-containing tails of **3**, **4** and **7** can be acceptors for the hydrogen bond from the amide sidechain of Gln92 (Figure 7); however, the steric hindrance of the ester moiety in **3** allows for forming only a weaker bond through the ethoxyl oxygen atom, while for **4** and **7**, a stronger bond is created with carbonyl or hydroxyl oxygen, which may explain the higher activity of these two compounds. The aliphatic substituent R^1^ of **5** and **6** and the rigid systems of **8**–**12** do not provide additional interactions beyond the hydrophobic contacts with Leu91Val121, Val131, and Leu135 and the weak aliphatic H–arene interactions.

The presence of an aminomethyl or thiomethyl linker in compounds **15**–**23** distances the 1,2,3-triazole ring from the pyridinesulfonamide core, which allows it to participate in the H–aryl interactions with the NH sidechain of Gln92. The carbonyl oxygen of the ester moiety of **15** and **16** may participate in hydrogen bonding with Gln67 (Figure 7).

Moderately active toward hCA IX compounds **19**–**20** form H–aryl interactions with the 1,2,3-triazole ring and NH sidechain of Gln92, while the phenyl ring located parallel above Leu91 forms weak but numerous H_—_arene interactions, whereas the lack of a methyl linker between the 1,2,3 triazole and 4-chlorophenyl group prevents this orientation toward Leu91 in the twice less active compounds **21** and **22**.

We did not observe any specific interactions that would explain the high activity of compounds **17**, **18**, and **23**, yet the docking algorithm proposed for compounds **4**, **16**, **17**, **18**, **21**, **22**, and **23** a reasonable second binding pose, analogous to that of hCA II, directed toward the hydrophilic part of the enzyme. The possibility of binding the inhibitor at two binding positions may have a significant contribution to activity and may be the reason for the high activity of these compounds. The phenomenon of different binding orientations in various carbonic anhydrase isoforms for *ortho*- or *meta*-substituted benzenesulfonamide derivatives has been shown by Virginija Dudutiene et al. [19,29]. Furthermore, the coexistence of both conformations within a single CA–ligand complex has also been noted, manifested by the presence of two alternate ligand positions in the single crystallographic structures (PDB models: 4PYY, 4Q08 (Figure 8)) [29].

#### 2.4.3. hCA XII

Again, docking to the hCA XII isoform for the majority of compounds, i.e., **5**, **6**, and **15**–**23**, can take two poses, being either directed toward the lipophilic or hydrophilic part of the enzyme. In Figure 8, two binding modes of compound **18**, compared to the two orientations observed for the reference compound in the X-ray structure of the hCA XII isoform, are presented. Interestingly, in this case, only the hydrophilic orientation of compounds **3**, **4**, and **7** was observed.

Since the binding pockets of isoforms II, IX, and XII do not differ substantially in structure on the hydrophilic side of the active cleft, the binding poses of compounds **3**–**7** and **15**–**23** toward hCA XII were analogous to those for hCA II and showed similar interactions with Trp5, Asn62, His64, and Thr200.

Yet, in orientation toward the lipophilic side, the two changes, compared to hCA IX, of the amino acids seem to have the biggest influence on the isoform selectivity of compounds **15**–**23**; these are changing Asn67 to Lys67 and Leu91 to Thr91. The presence of the Thr91 hydroxyl group allows compounds **17**–**23** to create new additional interactions, making them more active toward the hCA XII isoform. On the other side, compounds **15** and **16** do not form a hydrogen bond with the longer sidechain Lys67, in comparison to the presence of such a bond with the shorter Asn67 in hCA IX. Therefore, it could be the reason for the decrease in activity for hCA XII.

Regarding compounds **8**–**12**, which again display orientation only toward the lipophilic side, the change of hCA IX’s Val131 to smaller Ala131 provides more space for a linear three-ring structure, allowing for better fitting into the CAXII pocket than in the case of hCA IX (Figure 9). The rigid linear structure of compounds **8**–**12** is even more difficult to fit into the smaller active site of hCA II, which explains their high selectivity toward isozyme hCA XII compared to hCA IX and hCA II. 

In conclusion, the docking calculation appears to correlate with the generally observed trend that the CAI’s orientation with the tail directed toward the hydrophilic side is characteristic of hCA II due to the steric hindrance caused by Phe131, and it is associated with a lower inhibitory potency against this isoform. In contrast, the orientation toward the hydrophobic side is associated with higher activity, owing to the potential for multiple hydrophobic interactions [19,29]. Also, such an orientation provides many hydrogen interactions that explain the structure–activity relationship in the studied group of compounds. The different specificity of compounds **5** and **6**, i.e., low activity against hCA IX relative to hCA II and hCA XII, may be a result of the predominant role of the entropic contribution to the Gibbs energy of binding rather than from direct interactions. The possible dominant role of entropy gain to inhibitor binding has been shown for *ortho*- and *meta*-substituted benzenesulfonamide CA inhibitors [30,31]. This gain in entropy may be correlated with the binding pose not observed for hCA IX, where the long, labile hydrophobic hexyl or 3-methylbutyl chain of **5** and **6** is directed toward the partially hydrophilic cavity of hCA II and hCA XII, causing the displacement of water molecules.

#### 2.4.4. Antiproliferative Screening

Compounds **3**–**12** and **15**–**20** were evaluated for their effects on the viability of three cancer cell lines including HCT-116 (colon cancer), MCF-7 (breast cancer), and HeLa (cervical cancer), as well as for the representative compounds for non-cancerous keratinocytes (HaCaT). The concentration required for 50% inhibition of cell viability IC_50_ was determined using an MTT assay after 72 h of incubation with the tested compounds. As a positive control, cisplatin, doxorubicin, and tamoxifen (specific for breast cancer treatment) were used. The IC_50_ values are expressed as the mean of at least three independent experiments. The results are shown in Table 2.

The majority of the tested compounds exhibit a moderate ability to inhibit the growth of the examined cell lines, with the most sensitive being the HCT-116 colon cancer cell line (IC_50_ ranging from 17 to 260 µM), the MCF-7 breast cancer cell line (15 to 300 µM), and the HeLa cervical cancer cell line (22 to 460 µM). In comparison, for the reference compound cisplatin, the IC_50_ values for the tested cell lines were 3.8 µM, 3.1 µM, and 2.2 µM, respectively. No correlation was observed between the inhibition of any of the tested isoforms of carbonic anhydrase and the activity toward cancer cell lines.

The compounds showing the highest activity against all three tested cancer cell lines were compound **12**, with IC_50_ values of 17, 15, and 22 µM, and compound **22**, with IC_50_ values of 28, 19, and 39 µM, respectively. Both compounds are the weakest hCA II, IX, and XII inhibitors, indicating a mechanism of cytostatic action different from carbonic anhydrase inhibition. Interestingly, 4-aryl-1*H*-1,2,3-triazol-1-yl derivatives **8**–**12**, which are relatively weak carbonic anhydrase inhibitors, demonstrate a clear structure–activity relationship in terms of inhibiting cell viability. The least active compound, **8**, has an unsubstituted phenyl ring, with activity increasing as more substituents are introduced, i.e., 4-OCH_3_/4-F < 2-Me-4-OCH_3_ < 2,4,5-triMe. This pattern is opposite to that observed for carbonic anhydrase inhibition, which may be beneficial in light of minimizing the side effects associated with hCA inhibition in future anticancer compounds derived from the structure of **12**.

## 3. Materials and Methods

### 3.1. Carbonic Anhydrase Inhibition Assay

An Applied Photophysics stopped-flow instrument was used for assaying the CA-catalyzed CO_2_ hydration activity as previously reported [32]:

Phenol red (at a concentration of 0.02 mM) was used as the indicator, working at the absorbance maximum of 557 nm, with 20 mM Hepes (pH 7.5) as buffer, and 20 mM Na_2_SO_4_ (for maintaining a constant ionic strength), following the initial rates of the CA-catalyzed CO_2_ hydration reaction for a period of 10–100 s. The CO_2_ concentrations ranged from 1.7 to 17 mM for the determination of the kinetic parameters and inhibition constants. For each inhibitor, at least six traces of the initial 5–10% of the reaction have been used for determining the initial velocity. The uncatalyzed rates were determined in the same manner and subtracted from the total observed rates. Stock solutions of inhibitors (0.1 mM) were prepared in distilled–deionized water and dilutions up to 0.01 nM were prepared thereafter with distilled–deionized water. Inhibitor and enzyme solutions were preincubated together for 15 min at room temperature prior to the assay, in order to allow for the formation of the E–I complex. The inhibition constants were obtained by non-linear least-squares methods using PRISM 3 and the Cheng–Prussoff equation and represent the mean from at least three different determinations.

### 3.2. Cell Culture and Viability Assay

All chemicals, if not stated otherwise, were obtained from Sigma–Aldrich (St. Louis, MO, USA). The MCF-7, HeLa, and HaCaT cell lines were purchased from Cell Lines Services (Eppelheim, Germany), and the HCT-116 cell line was purchased from ATCC (ATCC-No: CCL-247). Cells were grown in Dulbecco’s modified Eagle’s medium (DMEM) supplemented with 10% fetal bovine serum, 2 mM glutamine, 100 units/mL penicillin, and 100 µg/mL streptomycin in an incubator (Heraceus, HeraCell) in a humidified atmosphere with 5% carbon dioxide at 37 °C.

The influence of the tested compounds on cell viability was examined using an MTT (3-(4,5-dimethylthiazol-2-yl)-2,5-diphenyltetrazolium bromide) assay. Cells at a density of 3 × 10^3^ cells/well were seeded in 96-well plates; then, the tested compounds in five concentrations of 1, 10 25, 50, and 100 µM, respectively, were added and incubated for 72 h. Next, MTT (0.5 mg/mL) was added to the medium, and cells were further incubated for 2 h at 37 °C. After that, cells were lysed with DMSO, and the absorbance of the formazan solution was measured at 550 nm with a plate reader (1420 multilabel counter, Victor, Jügesheim, Germany). The presented IC_50_ values were calculated as the mean ± SD of three independent experiments.

### 3.3. X-Ray Structure Determination

Diffraction intensity data for **8** and **18** were collected on an IPDS 2T dual-beam diffractometer (STOE & Cie GmbH, Darmstadt, Germany) at 120.0 (2) K with CuKa radiation of a microfocus X-ray source (GeniX 3D Mo High Flux, Xenocs, Sassenage, France, 50 kV, 0.6 mA, and *λ* = 1.54186 Å). Investigated crystals were thermostated under a nitrogen stream at 120 K using the CryoStream-800 device (Oxford CryoSystem, Oxford, UK) during the entire experiment.

Data collection and data reduction were controlled by using the X-Area 1.75 program (STOE). The numerical absorption correction was performed by integration. The structure was solved using intrinsic phasing implemented in SHELXT and refined anisotropically using the program packages Olex2 [33] and SHELX-2015 [34,35]. The positions of the C–H hydrogen atoms were calculated geometrically, and then the isotropic model was applied. The coordinates of H-atoms were refined as riding on their parent atoms with the usual restraints. The positions of H-atoms in -NH_2_ groups were found in the Fourier electron density map and refined unrestricted.

CCDC 2339695-2339696 contains the supplementary crystallographic data for this paper. The data can be obtained free of charge from The Cambridge Crystallographic Data Centre via www.ccdc.cam.ac.uk/structures (accessed on 14 April 2025).

### 3.4. Synthesis

The following instruments and parameters were used: for melting points, a Boetius HMK melting point apparatus (Reichert, Vienna, Austria) was used; IR spectra were measured on a Thermo Mattson Satellite FTIR spectrometer (Waltham, MA, USA) in KBr pellets, with an absorption range of 400–4000 cm^−1^. ^1^H-NMR and ^13^C-NMR spectra (Spectrums S1–S40) were recorded on a Varian Unity Plus 500 apparatus or on a Bruker Ascend 600 spectrometer (Bruker, Billerica, MS, USA); the chemical shifts are expressed as δ values relative to Me_4_Si (TMS).

The purity of compounds was analyzed by RP-HPLC on a Shimadzu (Model LC-10AD) HPLC system (Kyoto, Japan); Column: Gemini 4.6 × 250 mm; C6-phenyl; 5 µm; 110 Å, Mobile Phase: A—grade water with 0.1% (*v*/*v*) trifluoroacetic acid, B—80% acetonitrile-water containing 0.08% (*v*/*v*) trifluoroacetic acid, linear gradient 5–100% B in 60 min, Flow Rate: 1 mL/min.

Elemental analyses (C, H, N) were performed using a PerkinElmer 2400 Series II CHN Elemental Analyzer (Waltham, MA, USA).

4-Azidopyridine-3-sulfonamide (**2**) was obtained according to [36]. 4-(Prop-2-yn-1-ylthio)pyridine-3-sulfonamide (**14**) was obtained from **1** through thiouronium chloride (**14a**) according to the method described previously [16].

#### 3.4.1. General Method for Obtaining 4-(4-Methyl-1*H*-1,2,3-triazol-1-yl)pyridine-3-Sulfonamides **3**–**12**

In a round bottom flask, the appropriate alkyne (1–3 mmol) was placed and dissolved in 4 mL of acetonitrile. Next, 0.2 mmol (0.02 g, 0.028 mL) of triethylamine and 0.2 mmol (0.04 g) of copper (I) iodide was added, and the mixture was put under a nitrogen atmosphere. After that, a solution of 0.75 mmol 4-azidopyridine-3-sulfonamide (0.149 g) in 8 mL of acetonitrile was slowly added (up to 1 h), and the reaction was stirred overnight for 16 h at room temperature. Then, the solvent was evaporated under reduced pressure, and the remaining solid was suspended in 100 mL of ethyl acetate and washed with 2 × 15 mL of 0.02 M tetrasodium EDTA solution, 10 mL of 5% acetic acid, 10 mL of water, and 10 mL of brine. The organic phase was evaporated under reduced pressure, affording a crude product that was purified as follows.

##### Ethyl 1-(3-Sulfamoylpyridin-4-yl)-1*H*-1,2,3-triazole-4-carboxylate (**3**)

Using 3 mmol (0.294 g) of ethyl propiolate, the title compound was obtained after crude product crystallization from ethanol.

Yield 0.103 g (50%), mp 192–193 °C dec.;

IR (KBr) ν_max_ 3358, 3253 3155 (NH), 3108 (C_Ar_-H), 2988, 2962 (C-H), 1719 (C=O), 1579, 1533, 1497 (C=C, C=O), 1356, 1174 (SO_2_), 1240 (C-O) cm^−1^;

^1^H NMR (DMSO-*d*_6_, 500 MHz) δ: 1.32 (t, *J* = 7.1 Hz 3H, CH_3_), 4.36 (q, *J* = 7.0 Hz, 2H, CH_2_), 7.86 (d, *J* = 4.9 Hz, 1H, H-5pyridine), 7.96 (s, 2H, SO_2_NH_2_), 9.05 (d, *J* = 5.3 Hz 1H, H-6pyridine), 9.18 (s, 1H, H-2 pyridine), 9.26 (s, 1H, H-5 triazole) ppm.

^13^C NMR (DMSO-*d*_6_, 125 MHz) δ: 14.64, 61.40, 123.12, 132.26, 134.39, 139.40, 140.34, 149.91, 155.11, 160.41 ppm.

Anal. C 40.40; H 3.73, N 23.56% calcd for C_10_H_11_N_5_O_4_S (297.29), C 40.17; H 3.46; N 23.54%.

##### 1-(3-Sulfamoylpyridin-4-yl)-1*H*-1,2,3-triazole-4-carboxamide (**4**)

Using 3 mmol (0.207 g) of propiolamide obtained according to [37], the title compound was obtained after crude product crystallization from 80% ethanol.

Yield 0.083 g (41%), mp 203–205 °C;

IR (KBr) ν_max_ 3635, 3443, 3369, 3271 (NH), 3160, 3128 (C_Ar_-H), 2788 (C-H), 1682 (C=O), 1614, 1578, 1560, 1497 (C=C, C=O), 1353, 1171 (SO_2_) cm^−1^;

^1^H NMR (DMSO-*d*_6_, 500 MHz) δ: 7.66 (s, 1H, NH) 7.84 (d, *J* = 5.4 Hz, 1H, H-5pyridine), 7.96 (s, 2H, SO_2_NH_2_), 8.07 (s, 1H, NH), 8.96 (s, 1H, H-5 triazole), 9.03 (d, *J* = 5.4 Hz 1H, H-6 pyridine), 9.25 (s, 1H, H-2pyridine) ppm.

^13^C NMR (DMSO-*d*_6_, 150 MHz) δ: 123.01, 129.87, 134.20, 140.65, 143.49, 149.91, 155.12, 161.47 ppm.

Anal. C 35.82; H 3.01; N 31.33% calcd for C_8_H_8_N_6_O_3_S (268.25), C 35.45; H 2.69; N 30.93%.

##### 4-(4-Hexyl-1*H*-1,2,3-triazol-1-yl)pyridine-3-sulfonamide (**5**)

Using 3 mmol (0.330 g) of oct-1-yne, the title compound was obtained after crude product purification with column chromatography on silica gel and dichloromethane:methanol 100:3 as the mobile phase.

Yield 0.123 g (53%), mp 114–115 °C;

IR (KBr) ν_max_ 3290, 3162 (N-H), 2958, 2920, 2854 (C-H), 1580, 1491, 1461 (C=C, C=N), 1357, 1164 (SO_2_) cm^−1^;

^1^H NMR (DMSO-*d*_6_, 500 MHz) δ: 0.88 (t, *J* = 7.1 Hz, 3H, CH_3_), 1.18–1.38 (m, 8H, alkyl), 1.67 (quint, 2H, CH_2_), 2.73 (t, *J* = 7.7 Hz, 2H, alkyl), 7.79 (d, *J* = 4.8 Hz, 1H, H-5 pyridine), 7.87 (s, 2H, SO_2_NH_2_), 8.38 (s, 1H, H-5 triazole), 9.00 (d, *J* = 5.3 Hz, 1H, H-6 pyridine), 9.25 (s, 1H, H-2 pyridine) ppm.

^13^C NMR (DMSO-*d*_6_, 150 MHz) δ: 14.42, 22.51, 25.57, 28.59, 29.14, 31.49, 122.13, 124.86, 133.56, 141.18, 147.89, 150.00, 155.07 ppm.

Anal. C 50.47; H 6.19; N 22.64% calcd for C_13_H_19_N_5_O_2_S (309.39), C 50.20; H 6.05; N 22.45%.

##### 4-(4-(3-Methylbutyl)-1*H*-1,2,3-triazol-1-yl)pyridine-3-sulfonamide (**6**)

Using 3 mmol (0.294 g) of 5-methylhex-1-yne, the title compound was obtained after crude product crystallization from ethanol.

Yield 0.110. g (50%), mp 156–158 °C;

IR (KBr) ν_max_ 3329, 3228, 3147 (N-H), 3096, 3070 (C=O), 2973, 2957, 2931, 2871 (C-H), 1574, 1553, 1492, 1403 (C=C, C=N), 1336, 1162 (SO_2_) cm^−1^;

^1^H NMR (DMSO-*d*_6_, 600 MHz) δ: 0.93 (d, *J* = 6.6 Hz, 6H, 2×CH_3_), 1.55–1.64 (m, 3H, CH_2_-CH_2_-CH), 2.74 (t, *J* = 7.7 Hz, 2H, CH2-CH2-CH), 7.79 (d, *J* = 5.5 Hz, 1H, H-5 pyridine), 7.87 (s, 2H, SO_2_NH_2_), 8.39 (s, 1H, H-5 triazole), 9.00 (d, *J* = 5.1 Hz, 1H, H-6 pyridine), 9.24 (s, 1H, H-2 pyridine) ppm.

^13^C NMR (DMSO-*d*_6_, 150 MHz) δ:22.77, 23.25, 27.39, 38.28, 122.12, 124.77, 133.53, 141.18, 148.02, 150.01, 155.07 ppm.

Anal. C 48.80; H 5.80; N 23.71% calcd for C_12_H_17_N_5_O_2_S (295.11), C 48.70; H 5.79; N 23.07%.

##### 4-(4-(2-Hydroxypropan-2-yl)-1*H*-1,2,3-triazol-1-yl)pyridine-3-sulfonamide (**7**)

Using 2.25 mmol (0.189 g) of 2-methylbut-3-yn-2-ol, the title compound was obtained after crude product purification with column chromatography on silica gel and chloroform:methanol 10:1 as the mobile phase.

Yield 0.095 g (45%), mp 162–166 °C;

IR (KBr) ν_max_ 3332 (OH), 3162 (NH), 3065, 3029 (C=O) 2979, 2962 (CH), 1584, 1497 (C=C, C=N), 1353, 1172 (SO_2_), 1053 (C-O) cm^−1^;

^1^H NMR (DMSO-*d*_6_, 500 MHz) δ: 1.52 (s, 6H, 2 × CH_3_), 5.31 (s, 1H, OH), 7.79 (d, *J* = 4.9, 1H, H-5 pyridine), 7.86 (s, 2H, SO_2_NH_2_), 8.43 (s, 1H, H-5 triazole), 8.98 (d, *J* = 4.9, 1H, H-6 pyridine), 9.23 (s, 1H, H-2 pyridine) ppm.

^13^C NMR (DMSO-*d*_6_, 150 MHz) δ: 31.10, 67.52, 122.03, 123.49, 133.32, 141.10, 150.02, 155.10, 156.78 ppm.

Anal. C 42.39; H 4.63; N 24.72% calcd for C_11_H_14_N_4_O_3_S (283.31), C 41.89; H 4.17; N 24.54%.

##### 4-(4-Phenyl-1*H*-1,2,3-triazol-1-yl)pyridine-3-sulfonamide (**8**)

Using 3 mmol (0.306 g) of ethynylbenzene, the title compound was obtained after crude product crystallization from ethanol.

Yield 0.146 g (65%), mp 196–197 °C;

IR (KBr) ν_max_ 3314, 3122 (NH), 3093, 3047 (C_Ar_-H), 1579 1499, 1479 (C=C, C=N), 1357, 1169 (SO_2_) cm^−1^;

^1^H NMR (DMSO-*d*_6_, 500 MHz) δ: 7.4 (t, *J* = 7.3, 1H, H-4 phenyl), 7.51 (t, 2H, H-3,5 phenyl), 7.89 (d, *J* = 5.4, 1H, H-5 pyridine), 7.93 (m, 4H, SO_2_NH_2_, H-2,6 phenyl), 9.05 (s, 1H, H-5 triazole), 9.06 (brs, 1H, H-6 pyridine), 9.28 (s, 1H, H-2 pyridine) ppm.

^13^C NMR (DMSO-*d*_6_, 125 MHz) δ: 122.72, 124.59, 126.15, 129.13, 129.76, 130.53, 134.22, 141.18, 147.31, 150.34, 155.42 ppm.

Anal. C 51.82; H 3.68; N 23.24%, calcd for C_14_H_12_N_4_O_2_S (301.32), C 51.77; H 3.53; N 22.67 (wczesniej 23.26)%.

##### 4-(4-(4-Fluorophenyl)-1*H*-1,2,3-triazol-1-yl)pyridine-3-sulfonamide (**9**)

Using 1 mmol (0.120 g) of 1-ethynyl-4-fluorobenzene, the title compound was obtained after the crude product was washed with diethyl ether crystallized from ethanol.

Yield 0.096 g (40%), mp 207–209 °C dec;

IR (KBr) ν_max_ 3353, 3141, 3117 (NH), 3043 (C_Ar_-H), 1580, 1562, 1490 (C=C, C=N), 1348, 1169 (SO_2_) cm^−1^;

^1^H NMR (DMSO-*d*_6_, 500 MHz) δ: 7.35 (t, *J* = 8.8 Hz, 2H, H-3,5 phenyl), 7.87 (d, *J* = 4.9 Hz 1H H-5 pyridine), 7.93 (s, 2H, SO_2_NH_2_), 7.97 (dd, *J^H-H^* = 8.8 Hz, *J^H-F^* = 5.3 Hz, 2H, H-2,4 phenyl), 9.05 (m, 2H, H-5 triazole, H-6 pyridine), 9.27 (s, 1H, H-2 pyridine) ppm.

^13^C NMR (DMSO-*d*_6_, 125 MHz) δ: 116.41–116.58 (d *J* = 22 Hz), 122.50, 124.32, 126.89–126.86 (d, *J* = 4 Hz) 127.97–128.04 (d, *J* = 8.6 Hz), 134.01, 140.89, 146.20, 150.13, 155.18, 161.54–163.48 (d, *J* = 245 Hz).

Anal. C 48.90; H 3.16; N 21.93% calcd for C_13_H_10_FN_5_O_2_S (319.31), C; H; N %.

##### 4-(4-(4-Methoxyphenyl)-1*H*-1,2,3-triazol-1-yl)pyridine-3-sulfonamide (**10**)

Using 1 mmol (0.132 g) of 1-ethynyl-4-methoxybenzene, the title compound was obtained after crude product crystallization from ethanol.

Yield 0.072 g (29%), mp 177–179 °C;

IR (KBr) ν_max_ (SO_2_) 3370, 3263, 3139 (NH), 3078 (C_Ar_-H), 2974, 2947, 2840 (C-H) 1618 1577 1489, 1468 (C=C, C=N), 1349, 1170 (SO_2_) 1252, 1025 9 (C-O) cm^−1^;

^1^H NMR (DMSO-*d*_6_, 600 MHz) δ: 2.77 (s, 3H, CH_3_), 7.09 (d, *J* = 8.8 Hz*,* 2H, H-3,5 Ph), 7.87–7.89 (m, 3H, H-2,6 phenyl, H-5 pyridine), 7.92 (s, 2H, SO_2_NH_2_), 8.96 (s, 1H, H-5 triazole), 9.06 (d, *J* = 5.1 Hz 1H, H-6 pyridine), 9.29 (s, 1H, H-2 pyridine) ppm.

^13^C NMR (DMSO-*d*_6_, 125 MHz) δ: 55.69, 114.95, 122.38, 122.82, 123.31, 127.36, 133.90, 141.03, 147.10 150.15, 155.20, 159.89 ppm.

Anal. C 50.75; H 3.95; N 21.14% calcd for C_14_H_13_N_5_O_3_S (331.35), C 50.75; H 3.61; N 20.96%.

##### 4-(4-(4-Methoxy-2-methylphenyl)-1*H*-1,2,3-triazol-1-yl)pyridine-3-sulfonamide (**11**)

Using 1 mmol (0.146 g) of 1-ethynyl-4-methoxy-2-methylbenzene, the title compound was obtained after crude product crystallization from ethanol.

Yield 0.170 g (45%), mp 175–177 °C dec.;

IR (KBr) ν_max_: 3287, 3156 (N-H), 2975, 2950, 2920 (C-H), 1613, 1578, 1488, 1457, (C=C, C=N), 1357, 1171 (SO_2_) cm^−1^;

^1^H NMR (DMSO-*d*_6_, 500 MHz) δ: 2.47 (s, 3H, CH_3_), 3.37 (s, 3H, O-CH_3_), 6.92–6.94 (m, 2H, H-3,5 phenyl), 7.78 (d, *J* = 8.3 Hz, 1H, H-6 phenyl), 7.92 (d, *J* = 4.9 Hz, 1H, H-5 pyridine), 7.96 (s, 2H, SO_2_NH_2_), 8.78 (s, 1H, H-5 triazole), 9.06 (d, *J* = 4.9 Hz, 1H, H-6 pyridine), 9.29 (s, 1H, H-2 pyridine) ppm.

^13^C NMR (DMSO-*d*_6_, 125 MHz) δ: 21.73, 55.58, 112.22, 116.62, 122.10, 122.39, 125.41, 130.15, 133.77, 137.48, 141.01, 146.19, 150.02, 155.13, 159.57 ppm.

Anal. C 52.16; H 4.38; N 20.28% calcd for C_15_H_15_N_5_O_3_S (345.38), C 51.85; H 4.29; N 19.78%.

##### 4-(4-(2,4,5-Trimethylphenyl)-1*H*-1,2,3-triazol-1-yl)pyridine-3-sulfonamide (**12**)

Using 1 mmol (0.146 g) of 1-ethynyl-2,4,5-trimethylbenzene, the title compound was obtained after crude product crystallization from acetic acid.

Yield 0.119 g (46%), mp 195–199 °C dec.;

IR (KBr) ν_max_ 3282, 3169 (N-H), 3084 (C_Ar_-H), 2964, 2919, 2863 (C-H), 1579, 1488, 1458 (C=C, C=N), 1344, 1166 (SO_2_) cm^−1^;

^1^H NMR (DMSO-*d*_6_, 600 MHz) δ: 2.25 (s, 3H, CH_3_), 2.27 (s, 3H, CH_3_), 2.42 (s, 3H, CH_3_), 7.12 (s, 1H, H-3 phenyl.), 7.63 (s, 1H, H-6 phenyl), 7.93 (d, *J* = 5.2 Hz, 1H, H-5 pyridine), 7.96 (s, 2H, SO_2_NH_2_), 8.80 (s, 1H, H-5 triazole), 9.06 (d, *J* = 5.2 Hz, 1H, H-6 pyridine), 9.29 (s, 1H, H-2 pyridine) ppm.

^13^C NMR (DMSO-*d*_6_, 150 MHz) δ: 19.39, 19.52, 20.96, 122.40, 125.73, 126.85, 129.81, 132.69, 132.86, 133.79, 134.24, 136.82, 141.02, 146.43, 150.03, 155.13 ppm.

Anal. C 55.96; H 4.99; N 20.39% calcd for C_16_H_17_N_5_O_2_S (343.40), C 54.98; H 4.90; N 19.17%.

#### 3.4.2. 4-((Prop-2-yn-1-yl)amino)pyridine-3-sulfonamide (13)

In a round bottom flask, 10 mmol (1.92 g) of 4-chloropyridine-3-sulfonamide (**1**) and 20 mmol (1.10 g, 1.28 mL) of propargylamine in 20 mL of anhydrous ethanol was refluxed for 10 h. After that, the reaction mixture was cooled down, and the precipitating product was filtered off and dried.

Yield 1.30 g (69%), mp 202–204 °C;

IR (KBr) ν_max_ 3398, 3318, 3255 (N-H), 2917 (C-H), 2117 (C_sp_-H), 1599, 1561, 1518 (C=C, C=N), 1365, 1138 (SO_2_) cm^−1^;

^1^H NMR (DMSO-*d*_6_, 500 MHz) δ: 3.24 (s, 1H, CH), 4.14 (d, *J* = 3,4 Hz, 2H, CH_2_), 6.75 (t, *J* = 5.2 Hz, 1H, NH), 6.80 (d, *J* = 5.9 Hz, 1H, H-5 pyridine), 7.61 (s, 2H, SO_2_NH_2_), 8.28 (d, *J* = 5.9 Hz, 1H, H-6 pyridine), 8.53 (s, 1H, H-2 pyridine) ppm.

^13^C NMR (DMSO-*d*_6_, 125 MHz) δ: 31.98, 74.73, 80.54, 107.49, 122.79, 148.77, 149.00 152.80 ppm.

Anal. C 45.49; H 4.29; N 19.89% calcd for C_8_H_9_N_3_O_2_S (211.24), C 43.98; H 3.96; N 19.48%.

#### 3.4.3. General Method for Obtaining 4-(((1-Substituted-1*H*-1,2,3-triazol-4-yl)methyl)amino/thio)pyridine-3-sulfonamides **15**–**23**

A total of 1 mmol (0.211 g) of 4-(2-propyn-1-yl-amino)pyridino-3-sulfonamide **13** was dissolved in 2 mL of DMSO (or 1 mmol (0.228 g) of 4-(2-propyn-1-yl-tio)pirydyno-3-sulfonamide **14** dissolved in 4 mL of DMSO); then, 2 mmol of appropriate azide was added. Next, a freshly prepared mixture of 0.1 mmol of CuSO_4_ × 4 H_2_O (0.028 g) and 0.2 mmol of sodium ascorbate (0.04 g) in 0.5 mL of H_2_O and 1 mL of DMSO was added. The reaction mixture was stirred for 3 to 48 h at room temperature; then, 1 mL of glacial acetic acid was added and stirred for 15 min, causing the oxidation of Cu^1+^ ions and dissolution of all mixture components. Then, the reaction mixture was added dropwise to 20 mL of water with dissolved 0.5 mmol of tetrasodium EDTA; the precipitate that formed was filtered off, washed with water, and crystalized from ethanol or acetonitrile (compound **17**).

##### Ethyl 2-(4-(((3-sulfamoylpyridin-4-yl)amino)methyl)-1*H*-1,2,3-triazol-1-yl)acetate (**15**)

Using 4-(2-propyn-1-yl-amino)pyridino-3-sulfonamide **13** and ethyl 2-azidoacetate (0.258 g) obtained according to [38] and stirring the mixture for 48 h, the title compound was obtained.

Yield 0.112 g (33%), mp 179–180 °C;

IR (KBr) ν_max_ 3373, 3302, 3146 (N-H), 2999, 2957, 2860 (C-H), 1749 (C=O), 1595, 1522 (C=C, C=N), 1334, 1149 (SO_2_) cm^−1^;

^1^H NMR (DMSO-*d*_6_, 600 MHz) δ: 1.21 (t, *J* = 7.2 Hz, 3H, CH_3_), 4.12 (q, *J* = 7.2 Hz, 2H, CH_2_), 4.60 (d, *J* = 5.8 Hz, 2H, N-CH_2_), 5.38 (s, 2H, CH_2_), 6.88 (d, *J* = 5.8 Hz, 1H, H-5pyridine), 6.99 (t, *J* = 5.7 Hz, 2H, NH), 7.57 (s, 2H, SO_2_NH_2_), 8.03 (s, 1H, 5-H triazole), 8.22 (d, *J* = 5.9 Hz, H-6 pyridine), 8.52 (s, 1H, H-2 pyridine) ppm.

^13^C NMR (DMSO-*d*_6_, 125 MHz) δ: 14.42, 38.06, 50.85, 61.94, 107.26, 122.50, 124.92, 144.45, 148.85, 149.44, 152.82, 167.70 ppm.

Anal. C 42.35; H 4.74; N 24.69%, calcd for C_12_H_16_N_6_O_4_S (340.36), C 42.17; H 4.56; N 24.39%.

##### Ethyl 2-(4-(((3-Sulfamoylpyridin-4-yl)thio)methyl)-1*H*-1,2,3-triazol-1-yl)acetate (**16**)

Using 4-(2-propyn-1-yl-tio)pirydyno-3-sulfonamide **14** and ethyl 2-azidoacetate (0.258 g) obtained according to [38] and stirring the mixture for 3 h, the title compound was obtained.

Yield 0.220 g (61%), mp 140–141 °C;

IR (KBr) ν_max_: 3352, 3245, 3139 (NH), 3103, 3085 (C_ar_-H), 3000, 2930, 2909 (C-H), 1737 (C=O), 1567, 1526 (C=C, C=N) 1347, 1165 (SO_2_), 1229 (C-O) cm^−1^;

^1^H NMR (DMSO-*d*_6_, 500 MHz) δ: 1.19 (t, *J* = 7.3 Hz, 3H, CH_3_), 4.15 (q, *J* = 7.0 Hz, 2H, CH_2_), 4.51 (s, 2H, S-CH_2_), 5.37 (s, 2H, CH_2_), 7.65 (s, 2H, SO_2_NH_2_), 7.68 (d, *J* = 5.9 Hz, 1H, H-5 pyridine), 8.13 (s, 1H, 5-H triazole), 8.53 (d, *J* = 5.4 Hz, H-6 pyridine), 8.80 (s, 1H, H-2 pyridine) ppm.

^13^C NMR (DMSO-*d*_6_, 125 MHz) δ:14.40, 25.93, 50.87, 61.95, 120.76, 125.75, 135.88, 142.31, 147.75, 148.16, 151.86, 167.64 ppm.

Anal. C 40.33; H 4.23; N 19.59% calcd for C_12_H_15_N_5_O_2_S_2_ (357.41), C 39.24; H 3.86; N 19.26%.

##### 4-(((1-Cyclohexyl-1*H*-1,2,3-triazol-4-yl)methyl)amino)pyridine-3-sulfonamide (**17**)

Using 4-(2-propyn-1-yl-amino)pyridino-3-sulfonamide **13** and azidocyclohexane (0.250 g) obtained according to [39] and stirring the mixture for 48 h, the title compound was obtained.

Yield 0.085. g (25%), mp 181–182 °C;

IR (KBr) ν_max_ 3414, 3351, 3149 (N-H), 2937, 2860 (C-H), 1596, 1503 (C=C, C=N), 1340, 1147 (SO_2_) cm^−1^;

^1^H NMR (DMSO-*d*_6_, 600 MHz) δ: 1.20–1.26 (m, 1H, cyclohexyl), 1.38–1.45 (m, 2H, cyclohexyl), 1.65–1.82 (m, 5H, cyclohexyl), 2.02–2.04 (m, 2H, cyclohexyl), 4.43–4.48 (m, 1H, cyclohexyl), 4.54 (d, *J* = 5.5 Hz, 2H, N-CH_2_), 6.89 (d, *J* = 6.3 Hz, 1H, H-5 pyridine), 6.92 (t, *J* = 5.5 Hz, 1H, NH), 7.56 (s, 2H, SO_2_NH_2_), 8.07 (s, 1H, H-5 triazole), 8.23 (d, *J* = 5.9 Hz, 1H H-6 pyridine), 8.52 (s, 1H, H-2 pyridine) ppm.

^13^C NMR (DMSO-*d*_6_, 125 MHz) δ: 25.05, 25.13, 33.32, 38.22, 59.49, 107.25, 121.50, 122.41, 143.90, 148.88, 149.46, 152.89 ppm.

Anal. C 49.98; H 5.99; N 24.98% calcd for C_14_H_20_N_6_O_2_S (336.41), C 49.90; H 5.97; N 24.91%.

##### 4-(((1-Cyclohexyl-1*H*-1,2,3-triazol-4-yl)methyl)thio)pyridine-3-sulfonamide (**18**)

Using 4-(2-propyn-1-yl-tio)pirydyno-3-sulfonamide **14** and azidocyclohexane (0.250 g) obtained according to [39] and stirring the mixture for 48 h, the title compound was obtained.

Yield 0.224 g (63%), mp 211–212 °C;

IR (KBr) ν_max_ 3364, 3141 (N-H), 3007 (C_Ar_-H), 2939, 2860 (C-H), 1572, 1451 (C=C, C=N), 1340, 1162 (SO_2_) cm^−1^;

^1^H NMR (DMSO-*d*_6_, 500 MHz) δ: 1.17–1.25 (m, 1H, cyclohexyl), 1.35–1.44 (m, 2H, cyclohexyl), 1.63–1.81 (m, 5H, cyclohexyl), 2.00–2.03 (m, 2H, cyclohexyl), 4.40–4.46 (m, 3H, H-1 cyclohexyl, SCH_2_), 7.64 (s, 2H, SO_2_NH_2_), 7.69 (d, *J* = 5.5 Hz, 1H, H-5 pyridine). 8.17 (s, 1H, H-5 triazole), 8.55 (d, *J* = 5.5 Hz, 1H H-6 pyridine), 8.80 (s, 1H, H-2 pyridine) ppm.

^13^C NMR (DMSO-*d*_6_, 125 MHz) δ:25.02, 25.09, 26.26, 33.25, 59.54, 120.84, 122.32, 135.86, 141.76, 147.71, 148.24, 151.90 ppm.

Anal. C 47.57; H 5.42; N 19.81% calcd for C_14_H_19_N_5_O_2_S_2_ (353.46), C 47.60; H 5.26; N 19.62%.

##### 4-(((1-Benzyl-1*H*-1,2,3-triazol-4-yl)methyl)amino)pyridine-3-sulfonamide (**19**)

Using 4-(2-propyn-1-yl-amino)pyridino-3-sulfonamide **13** and (azidomethyl)benzene (0.266 g) obtained according to [40] and stirring the mixture for 3 h, the title compound was obtained.

Yield 0.200 g (58%), mp 184–185 °C;

IR (KBr) ν_max_ 3380, 3125 (N-H), 3088, 3062 (C_Ar_-H), 2988, 2913 (C-H), 1597, 1565, 1508 (C=C, C=N), 1324, 1148 (SO_2_) cm^−1^;

^1^H NMR (DMSO-*d*_6_, 500 MHz) δ: 4.55 (d, *J* = 5.5 Hz, 2H, N-CH_2_), 5.57 (s, 2H, CH_2_), 6.86 (d, *J* = 5.9 Hz, 1H, H-5 pyridine), 6.93 (t, 1H, NH),7.27–7.38 (m, 5H, phenyl), 7.52 (s, 2H SO_2_NH_2_), 8.06 (s, 1H, H-5 triazole), 8.21 (br.s., 1H, H-6 pyridine), 8.50 (s, 1H, H-2 pyridine) ppm.

^13^C NMR (DMSO-*d*_6_, 125 MHz) δ: 38.14, 53.25, 107.31,123.63, 128.36, 128.60, 129.22, 136.53, 144.68, 148.88, 149.41, 152.83 ppm.

Anal. C 52.31; H 4.68; N 24.40% calcd for C_15_H_16_N_6_O_2_S (344.39), C 52.42; H 4.32; N 24.32%.

##### 4-(((1-Benzyl-1*H*-1,2,3-triazol-4-yl)methyl)thio)pyridine-3-sulfonamide (**20**)

Using 4-(2-propyn-1-yl-tio)pirydyno-3-sulfonamide **14** and (azidomethyl)benzene (0.266 g) obtained according to [40] and stirring the mixture for 3 h, the title compound was obtained.

Yield 0.292 g (81%), mp 184–185 °C;

IR (KBr) ν_max_ 3125 (NH), 3088, 3055 (C_Ar_-H) 2999, 2982 (C-H), 1571 1497, 1403 (C=C, C=N), 1349, 1163 (SO_2_) cm^−1^;

^1^H NMR (DMSO-*d*_6_, 600 MHz) δ: 4.49 (s, 2H, S-CH_2_), 5.59 (s, 2H, CH_2_), 7.28–7.37 (m, 5H, phenyl), 7.67 (S, 2H SO_2_NH_2_), 7.69 (d, *J* = 6.8 Hz, 1H, H-5 pyridine), 8.20 (s, 1H, H-5 triazole), 8.55 (d, *J*= 5.2 Hz, 1H, H-6 pyridine), 8.82 (s, 1H, H-2 pyridine) ppm.

^13^C NMR (DMSO-*d*_6_, 125 MHz) δ: 26.08, 53.33, 120.86, 124.47, 128.34, 128.64, 129.25, 135.92, 136.42, 142.60, 147.75, 148.16, 151.88 ppm.

Anal. C 49.84; H 4.18; N 19.38% calcd for C_15_H_15_N_5_O_2_S_2_ (361.44), C 49.19; H 4.01; N 19.22%.

##### 4-(((1-(4-Chlorophenyl)-1*H*-1,2,3-triazol-4-yl)methyl)amino)pyridine-3-sulfonamide (**21**)

Using 4-(2-propyn-1-yl-amino)pyridino-3-sulfonamide **13** and 1-azido-4-chlorobenzene (0.306 g) obtained according to [41] and stirring the mixture for 24 h, the title compound was obtained.

Yield 0.215 g (59%), mp 226–228 °C;

IR (KBr) ν_max_ 3352, 3299, 3161 (NH), 3100, 3084 (C_Ar_-H) 2956, 2866 (C-H), 1602 1523 1499 (C=C, C=N), 1335, 1148 (SO_2_) cm^−1^;

^1^H NMR (DMSO-*d*_6_, 600 MHz) δ: 4.68 (d, *J* = 5.8 Hz, 2H, N-CH_2_), 6.94 (d, *J* = 6.2 Hz, 1H, H-5 pyridine), 7.01 (t, 1H, NH), 7.57 (s, 2H, SO_2_NH_2_), 7.67 (d, *J* = 8.8 Hz, 2H, H-3,5 phenyl), 7.93 (d, *J* = 8.8 Hz, 2H, H-2,6 phenyl), 8.26 (d, *J* = 8.8 Hz, 1H, H-6 pyridine), 8.54 (s, 1H, H-5 triazole), 8.76 (s, 1H, H-2 pyridine) ppm.

^13^C NMR (DMSO-*d*_6_, 150 MHz) δ: 38.03, 107.32, 121.95, 122.24, 122.54, 130.36, 133.46, 135.82, 145.74, 148.91, 149.43, 152.98 ppm.

Anal. C 46.09; H 3.59; N 23.04%, calcd for C_14_H_13_ClN_6_O_2_S (364.81), C 46.10; H 3.83; N 22.29%.

##### 4-(((1-(4-Chlorophenyl)-1*H*-1,2,3-triazol-4-yl)methyl)thio)pyridine-3-sulfonamide (**22**)

Using 4-(2-propyn-1-yl-tio)pirydyno-3-sulfonamide **14** and 1-azido-4-chlorobenzene (0.306 g) obtained according to [41] and stirring the mixture for 24 h, the title compound was obtained.

Yield 0.293 g (77%), mp 235–237 °C;

IR (KBr) ν_max_ 3228, 3150 (NH), 3088, 3068 (C_Ar_-H), 2989 (C-H), 1567, 1498 (C=C, C=N), 1329, 1162 (SO_2_) cm^−1^;

^1^H NMR (DMSO-*d*_6_, 500 MHz) δ: 4.58 (s, 2H, S-CH_2_), 7.65–7.66 (m, 4H, H-3,5 phenyl, SO_2_NH_2_), 7.73 (d, *J* = 5.5 Hz, 1H, H-5 pyridine), 7.90 (d, 2H, H-2,6 phenyl), 8.56 (d, *J*= 5.5 Hz, 1H, H-6 pyridine), 8.81 (s, 1H, H-5 triazole), 8.82 (s, 1H, H-2 pyridine) ppm.

^13^C NMR (DMSO-*d*_6_, 125 MHz) δ: 25.98, 120.98, 122.22, 122.66,130.33, 133.50, 135.70, 136.01, 143.94, 147.78, 147.81, 151.99 ppm.

Anal. C 44.03; H 3.17; N 18.34% calcd for C_14_H_12_ClN_5_O_2_S (381.86), C 43.93; H 2.95; N 18.03%.

##### 4-(((1-((1,3-Dioxolan-2-yl)methyl)-1*H*-1,2,3-triazol-4-yl)methyl)thio)pyridine-3-sulfonamide (**23**)

Using 4-(2-propyn-1-yl-tio)pirydyno-3-sulfonamide **14** and 2-(azidomethyl)-1,3-dioxolane (0.268 g) obtained according to [42] and stirring the mixture for 24 h, the title compound was obtained.

Yield 0.230 g (65%), mp 137–138 °C;

IR (KBr) ν_max_ 3355, 3159 (NH), 3088, 3009 (C_Ar_-H), 2902 (CH), 1646, 1573 (C=C, C=N), 1054 (C-O), 1338, 1155 (SO_2_) cm^−1^;

^1^H NMR (DMSO-*d*_6_, 500 MHz) δ: 3.70–3.79 (m, 4H, 2xCH_2_), 4.47 (s, 2H, S-CH_2_), 4.52 (d, *J* = 3.8 Hz*,* 2H, CH_2_), 5.18 (t, *J* = 3.8 Hz, 1H, CH), 7.64 (s, 2H, SO_2_NH_2_), 7.68 (d, *J* = 5.4 Hz, 1H, H-5 pyridine), 8.05 (s,1H, H-5 triazole), 8.54 (d, *J* = 5.4 Hz, 1H, H-6 pyridine), 8.79 (s, 1H, H-2 pyridine) ppm.

^13^C NMR (DMSO-*d*_6_, 125 MHz) δ: 25.95, 51.95, 65.07, 100.97, 120.83, 125.21, 135.93, 142.24, 147.72, 148.11, 151.83 ppm.

Anal. C 40.33; H 4.23; N 19.59% calcd for C_12_H_15_N_15_O_4_S_2_ (357.41), C 39.38; H 4.46; N 18.42%.

### 3.5. Docking Studies

The 3D structures of investigated sulfonamides **3**–**23** in anionic form were sketched and optimized in an Amber10:EHT forcefield using Molecular Operating Environment (MOE), 2022.02 software. As target proteins, X-ray structures of human carbonic anhydrase hCA II (PDB code 6RH4), hCA IX (PDB code 5DVX), and hCA XII (PDB code 6QNG) obtained from Protein Data Bank were used. Particular PDB structures were selected by cross-validation (on 10 ligands for hCA II, 14 ligands for hCA IX, and 27 ligands for hCA XII) from all those available in the rcsb database (https://www.rcsb.org/ (accessed on 30 December 2024)) for hCA IX and XII and from 7 preselected models for hCA II. The protein model for which the mean RMSD of the highest-scoring docked poses was the lowest was selected.

Proteins structures were prepared using the MOE “QuickPrep” option, including the correction of structural errors, protonation, calculation of partial charges, tethering active site atoms, and fixing atoms farther than 8 Å from the original ligand, completed with energy minimization in the Amber10:EHT forcefield. For the docking calculation, all water molecules were omitted. Please note that the protonation state of the enzyme active center, i.e., geometry and tautomeric form of histidine residues and glutamine, had to be prepared manually.

The two-step docking procedure implemented in MOE was used: first, using the “Triangle Matcher” placement method and London dG scoring function, 250 best-fitted ligand poses were selected and were further subjected to refinement, which optimized the conformation using molecular mechanics and rescored them with the GBVI/WSA dG scoring function, leaving the 10 top-rated poses. During refinement, protein was treated as rigid for the hCA XII model, while for hCA II and hCA IX, induced fitting with tethered sidechains was applied. All resulting poses were analyzed based on the scoring value, interaction similarity, and similarity to known complexes (poses with the arrangement of the aryl sulfonamide fragment significantly different from known crystallographic complexes were rejected). The visualization and analysis of the obtained results were performed with MOE software.

## 4. Conclusions

New pyridine-3-sulfonamide derivatives **3**–**12** and **15**–**23** were successfully synthesized using copper (I)-catalyzed CuAAC cycloaddition reactions, both when the azide (comp. **2**) or alkyne group (comp. **13** and **14**) was present as a substituent on the pyridine-3-sulfonamide scaffold. The structure of the compounds and the selectivity of the Cu(I)-catalyzed reaction were confirmed crystallographically. The compounds exhibited lower activity against carbonic anhydrase isoforms compared to our previously synthesized pyridine-3-sulfonamide derivatives and showed general higher selectivity toward isoform XII of carbonic anhydrase over isoforms IX and II. Molecular docking suggests that similar to *ortho*-substituted arylsulfonamides, the new compounds, depending on the CA isoform, are capable of adopting a pose directed with a tail either toward the hydrophilic or toward lipophilic sides of the active site. As expected, two types of ligand orientations that can sometimes co-exist within one isoform provide a number of specific interactions and result in the high selectivity of some compounds. The general structure–activity relationship shows that compounds with a flexible sidechain exhibited greater activity compared to those with a rigid structure, particularly compounds **5** and **6** versus **8** or **19**–**20** versus **21**–**22**. On the other hand, the rigid linear structure of compounds **8**–**12** is probably the key to their high selectivity toward hCA XII. Compounds **12** and **22** demonstrated the highest cytostatic activity against HCT-116, HeLa, and MCF-7 cell lines, while not being potent CA inhibitors. This could be advantageous in further studies focused on cytostatic activity unrelated to CA inhibition, potentially minimizing side effects.

## Data Availability

Spectra of new compounds are available as Appendix A. Other data, such as the results of molecular calculations or compound samples are available upon contact with the corresponding author.

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
