# Peer review of "4-Substituted Pyridine-3-Sulfonamides as Carbonic Anhydrase Inhibitors Modified by Click Tailing: Synthesis, Activity, and Docking Studies"

_ijms, 2025, doi:10.3390/ijms26083817_

Round 1
Reviewer 1 Report
Comments and Suggestions for Authors
International Journal of Molecular Science
4-Substituted pyridine-3-sulfonamides as carbonic anhydrase inhibitors modified by click tailing: synthesis, activity, and docking studies
In the work entitled “4-Substituted pyridine-3-sulfonamides as carbonic anhydrase inhibitors modified by click tailing: synthesis, activity, and docking studies” the authors describe the synthesis of new pyridine-3-sulfonamide derivatives as new selective inhibitors of human carbonic anhydrase (hCA). The compounds were obtained through a click CuAAc reaction and were properly characterized, proving their structure. Moreover, docking studies and cytostatic activity were carried out providing promising results. Based on this, I consider it appropriate to publish this manuscript in International Journal of Molecular Science after the following considerations.
- Carefully review the writings which should be in italics such as "ortho, in situ".
- In my opinion, in the description of the chemical synthesis it should be specified how many hours are meant by the term "overnight".
- In Scheme 1, authors write “acetonitrile” the correct is “acetonitrile”.
- I think it is better to specify in the legend of figure 4 that the images refer to molecule 8.
- The “Table 1” in line 433 should be “Table 2”.
- In both tables, Table 1 and 2, there are values ​​described with different colours, I think it is appropriate to specify the reason for this type of description, what the different colour depends on.
- In the end all the new synthesized compounds have a lower inhibitory activity than the clinically already used molecule, what is then the advantage of having this new battery of compounds available?
Author Response
In the work entitled “4-Substituted pyridine-3-sulfonamides as carbonic anhydrase inhibitors modified by click tailing: synthesis, activity, and docking studies” the authors describe the synthesis of new pyridine-3-sulfonamide derivatives as new selective inhibitors of human carbonic anhydrase (hCA). The compounds were obtained through a click CuAAc reaction and were properly characterized, proving their structure. Moreover, docking studies and cytostatic activity were carried out providing promising results. Based on this, I consider it appropriate to publish this manuscript in International Journal of Molecular Science after the following considerations.
-
Carefully review the writings which should be in italics such as "ortho, in situ".
We have corrected it, trying to follow IUPAC guidelines.
-
In my opinion, in the description of the chemical synthesis it should be specified how many hours are meant by the term "overnight".
By overnight we meant a time of about 16 hours. This has been corrected in the diagram in the experimental description..
-
In Scheme 1, authors write “acetonitrile” the correct is “acetonitrile”.
Corrected
-
I think it is better to specify in the legend of figure 4 that the images refer to molecule 8.
This information has been added. Additionally, the figure 4 caption “View of the four symmetry independent molecules presenting their conformation and internal N-H…N bonding in A and B (drawn as dashed cyan lines).“ has been changed to a clearer “View of the four independent conformers (A B C D) of comp 8 presenting their internal N-H…N bonding in A and B (drawn as dashed cyan lines).” and the figure itself, which was a table, has been replaced with a better quality image.
-
The “Table 1” in line 433 should be “Table 2”.
Corrected
-
In both tables, Table 1 and 2, there are values ​​described with different colours, I think it is appropriate to specify the reason for this type of description, what the different colour depends on.
Heatmap colouring was used to facilitate visual analysis of data from tables, e.g. comparison of activity order for different hCA isoforms. Colours change in a gradient from red (lowest activity) to green (highest activity). An appropriate legend was added in the table footer.
-
In the end all the new synthesized compounds have a lower inhibitory activity than the clinically already used molecule, what is then the advantage of having this new battery of compounds available?
Although our compounds showed generally lower activity than the reference acetazolamide, this difference was not always overwhelmingly large: 137.5 nM (comp 4) vs 25 nM AAZ for hCA9 or 91 (comp 6) vs 5.7 AAZ for hCA12. But most importantly, compared to the low selectivity of clinically used acetazolamide, the described compounds are characterized by even 10-20-fold selectivity towards isoforms 9 and 12, which is an advantage and a goal we strive for in the potential use of CA inhibitors in cancer therapy. Additionally, the compounds showed very diverse activity, which is firstly interesting from the point of view of the structure-activity relationship and secondly it indicates a wide potential to influence activity by using this type of modification ("click tailing") of the basic pyridine 3-sulfonamide pharmacophore.
Reviewer 2 Report
Comments and Suggestions for Authors
I extend my sincere appreciation to the authors for their valuable work. I would, however, like to raise the following points for consideration:
1- More highlight is needed on the hCA IX and XII inhibition significance in cancer treatment.
2- Why the 4-position in pyridine was specifically targeted for substitution ?
3- Regarding compounds synthesized with low yields, did the author try more optimizations for the reaction conditions ?
4- I hope more docking explanation for compounds with the best Ki values is needed.
5- Why no obvious correlation between carbonic anhydrase inhibition and cytotoxic effects?
6- I see that MOE was used for docking simulations, why you didn't adopt induced fit docking technique ?
7- I think use of IC50 in abstract should be corrected ?
Comments on the Quality of English Language
The quality of the English language is generally adequate; however, further revision would enhance clarity and readability
Author Response
Extend my sincere appreciation to the authors for their valuable work. I would, however, like to raise the following points for consideration:
-
More highlight is needed on the hCA IX and XII inhibition significance in cancer treatment.
We have added the appropriate information to the introduction. Lines 53-69
-
Why the 4-position in pyridine was specifically targeted for substitution ?
Most studies on anhydrase inhibitors focus on arylsulfonamide-based inhibitors substituted in the “para” position to the sulfonamide group. Such a substitution, due to the vertical structure of the active site, provides a large degree of freedom in the interaction of the tail with the enzyme surface. We assumed that the substituent closer to the zinc-binding group (in the “ortho” position), i.e. positions 2 or 4 for pyridine-3-sulfonamide scaffold (of which position 4 is more optimal from a synthetic point of view ), would provide forced contact and thus more interaction with the side walls of the active site, which may result in higher selectivity between similarly constructed isoforms. Examples of ortho-substituted benzenesulfonamide derivatives with very good selectivity are shown in Figure 2. We describe this in the theoretical introduction, lines 86 to 113
-
Regarding compounds synthesized with low yields, did the author try more optimizations for the reaction conditions ?
We did optimizations of the synthesis, in terms of the solvent and reaction conditions, but we were unable to significantly increase the yield. The low yield resulted rather from the strong complexation of copper ions by the pyridine ring. The complexes that were formed were difficult to dissolve, which prevented the use of chromatographic methods, and what is more, they could recreated during crystallization, which had to be repeated. However, even trace amounts of Cu ions had a significant effect on the NMR spectra, causing the loss of multiplicity pattern and broadening of the protons resonance signals and the loss of the carbon signals of the pyridine ring. Therefore, and ensuring the quality of biological studies (Cu ions could affect the spectrophotometric measurement), we placed purity above yield.
-
I hope more docking explanation for compounds with the best Ki values is needed.
Unfortunately, no scoring function allowed us to distinguish the best compounds with very low Ki values ​​from less active, but still binding, compounds. This is a problem we observed in most publications describing docking to anhydrase. Therefore, we do not compare the scoring values ​​of the compounds and in the validation we focused on reproducing the true binding pose (low RMSD ). Where it was possible, we discussed the influence of potential interactions with the protein on the activity, such as the discussion of the influence of the differences in the carbonyl group of compounds 3, 4 and 7 in part 2.4.2. Unfortunately, for compounds 4 and 5 with the most interesting and highest activity towards hCA2 and hCA12 ass well as 18 or 23, it was not possible to observe interactions explaining their high activity other than lipophilic ones. At the end of section 2.4.3 we refer to the studies of A. Zubrienė, et al. and A. Smirnov et al. who write about the predominant role of the entropic contribution of the Gibbs Energy in the term of binding to the CA active site. And we suspect a similar mechanism in the case of our most active compounds what we write there too. I think that from the docking calculations performed, it is not possible to draw more conclusions that would be interesting from the reader's point of view. In turn, further extension of the research using computational methods, which we are working on, would go beyond the scope of this single publication..
5- Why no obvious correlation between carbonic anhydrase inhibition and cytotoxic effects?
This is most likely due to the fact of different conditions of in vitro cell culture growth, than those found in hypoxic tumor tissues and related to this overexpression of tumor associated hCA isozymes.
-
- I see that MOE was used for docking simulations, why you didn't adopt induced fit docking technique ?
We used the induced fit method for the models of anhydrase 2 and 9. It is writen, maybe too briefly, in experimental part 4.5 Docking studies.
When optimizing the docking algorithm with cross-validation, we were guided by the lower RMSD values ​​obtained for known crystallographic complexes. In the case of hCA2 and 9, the use of induced fit with tethers significantly improved the prediction of poses (on the other hand, the use of completely “free to move” amino acid residues led to a overfitting ligands into the active site). For hCA12, the use of induced fit did not cause a significant decrease in the RMSD value but significantly extended computational time.
7- I think use of IC50 in abstract should be corrected ?
Yes, we actually wrote IC50 instead of KI. Thank you for pointing this out.
Reviewer 3 Report
Comments and Suggestions for Authors
The paper describes several new triazole-quinoline based high nanomolar inhibitors of carbonic anhydrase. Some display good inhibition and enable SAR study. The authors also perform docking computations, which the reviewer will not refer to. Finally the authors study antiproliferative activity of their products and obtain some measurable activity against some cell lines.
The reviewer notes that some similar synthetic difficulties faced by the authors due to the complexation of copper have been resolved in the preceding literature. The authors should refrain from using the term "anticancer effect" for studies other than in vivo experiments.
The paper is well written and deserves publication with insignificant corrections:
1) Adjust the layout of Figures to match the article print width
2) Improve chemical nomenclature "4-Azidepyridine-3-sulfonamide" (ln.122), Ph when referring to C6H4 ... C6H2 (Schemes 1-2)
Author Response
The paper describes several new triazole-quinoline based high nanomolar inhibitors of carbonic anhydrase. Some display good inhibition and enable SAR study. The authors also perform docking computations, which the reviewer will not refer to. Finally the authors study antiproliferative activity of their products and obtain some measurable activity against some cell lines.
The reviewer notes that some similar synthetic difficulties faced by the authors due to the complexation of copper have been resolved in the preceding literature.
The authors should refrain from using the term "anticancer effect" for studies other than in vivo experiments.
The term "anticancer" has indeed been overused. The phrases "anticancer effect" "anticancer activity" etc. have been changed where it apply to the described research, to more appropriate ones such as cytotoxicity or antiproliferative or viability of cancer cell lines..
The paper is well written and deserves publication with insignificant corrections:
-
Adjust the layout of Figures to match the article print width
Comparing other articles from IJMS in pdf version we see that very often figures and tables are wider than the column with text. Files with figures are attached so that the editor can arrange them appropriately.
2) Improve chemical nomenclature "4-Azidepyridine-3-sulfonamide" (ln.122), Ph when referring to C6H4 ... C6H2 (Schemes 1-2)
The errors have been corrected.